# Professional Nurses' Experiences of Student Nurses' Absenteeism during Psychiatric Clinical Placement in Limpopo Province, South Africa: A Qualitative Study

Thingahangwi C. Masutha *, Mary Maluleke, Ndidzulafhi S. Raliphaswa, Mphedziseni E. Rangwaneni, Takalani E. Thabathi, Ndivhaleni R. Lavhelani and Duppy Manyuma

Department of Advanced Nursing, University of Venda, Private Bag X5050, Thohoyandou 0950, South Africa
* Correspondence: thingahangwi.masutha@univen.ac.za

**Abstract:** Psychiatric clinical placement is as essential as other placements in any discipline in nursing education as it allows student nurses to correlate theory to practice. Several research studies have been conducted on nurses' experiences of the clinical supervision of students, but absenteeism is still a challenge globally. In this study, professional nurses' experiences of student nurses' absenteeism from psychiatric clinical placements in Limpopo Province, South Africa, were examined. Three designated specialty mental institutions in Limpopo Province were the sites of the study. Explorative, descriptive, and contextual designs were used in a qualitative approach. A qualitative approach was used because the study sought professional nurses to narrate their experiences through in-depth, semi-structured interviews, which could only be achieved through a qualitative approach. A total of eleven professional nurses, four from two specialized psychiatric hospitals and three from one specialized psychiatric hospital, were purposively sampled and interviewed. These nurses participated in in-depth interviews that were used to gather data, which were then thematically analyzed. One theme and three sub-themes on professional nurses' experiences of the absenteeism of student nurses during psychiatric clinical placement were revealed. The theme was student factors leading to absenteeism, while the sub-themes were fear of mental health care users, negative attitudes towards psychiatric nursing science, and students' lifestyles. Based on the findings, student nurses' absenteeism can be caused by many factors. A qualitative study should be conducted to seek strategies to mitigate the absenteeism of student nurses during clinical placement in the psychiatric hospitals of Limpopo Province.

**Keywords:** absenteeism; clinical placement; experiences; interviews; professional nurses; student nurses

## 1. Introduction

Regardless of the cause, absenteeism is defined as failing to attend scheduled lessons and training sessions [1]. Absenteeism might be characterized as not attending class or not following a course without having a legal reason [2]. Studies reveal that inadequate study has been conducted on the experiences of clinical nurses who are engaged in educating nursing students outside of their regular working hours, even though it is well-known how important nurses' roles are in giving clinical training to nursing students [3]. However, clinical education is an essential part of healthcare professionals' curricula around the world. Under the guidance of clinical supervisors, it gives students a chance to put the theory they learn in the classroom into practice [4]. To ensure that new nurses are competent, nursing education strongly emphasizes student learning during clinical placement [5]. In Norway, clinical experience that includes classroom instruction in a simulation lab constitutes 50% of the nursing bachelor's degree program. Learning practical skills is essential in both learning contexts [6]. Nursing students practice various practical skills during clinical placement while nurses supervise them [5].

The five main components of clinical education are training institutions, clinical departments, supervisors, students, and patients, according to Bwanga and Chanda [4]. Firstly, training institutions ensure supervisors are supported and students are sufficiently prepared for placements. In addition, clinical departments must have enough faculty and instructional materials so that students can follow up on their experiences. Thirdly, supervisors should oversee resources, instruct pupils, serve as role models, provide them with feedback, evaluate their performance, and assist those who are having difficulty. Fourthly, students should actively identify their learning requirements and have a positive outlook on learning [7]. Finally, patients need to be ready to take part in medical professionals' training. Student nurses are required to complete 80% of the total hours in mental health nursing science for clinical exposure, as stated by the South African Nursing Council [8]. Nursing students are required by the Irish Nursing Council [9] to complete 100% of their required clinical hours each year. Finland, on the other hand, requires 95% attendance to fulfil the course requirements [10]. Similarly, in order to take an exam for a particular course at the Nigerian University of Abadan, nursing students must have 75% attendance [11].

Due to a high absence rate among level III and IV student nurses observed during mental health nursing science clinical placement at Limpopo College of Nursing, there has been a lack of clinical hours during the accompanying of students for the past few years. The evidence for this claim is in [12]. According to Magobolo and Dube [10], a lack of mental health nursing clinical hours was caused by 97.3% of nursing students skipping their clinical rotations. According to Chukwu et al. [11], nursing students' absenteeism from clinical settings and lectures is a serious issue in Nigeria, where the vast majority of students often skip class. In a similar manner, Ref. [1] suggests that absenteeism in universities and colleges is problematic. According to a survey conducted at Ethiopian universities, many higher education institutions have tight rules requiring students to attend lectures, laboratory, and practical sessions. Absenteeism in those universities persists in spite of those policies. On the other hand, Ref. [13] claims that since 1998 student absenteeism in the technical and vocational education and training sector has increased significantly in the majority of South African provinces. Limpopo Province has only three specialized psychiatric hospitals where students from different institutions and disciplines are placed for psychiatric clinical learning opportunities. A multidisciplinary team of medical students, physiotherapy students, and other nursing students from the universities was also assigned to psychiatric clinical learning opportunities, in addition to nursing students from the college. As a result, there was overcrowding, which made it difficult for licensed nurses to supervise students and for nursing students to accomplish their clinical requirements. There was also poor integration of theory and practice. Student nurses' absences caused a shortfall of clinical hours, which led to an extension of their training.

Because a student is not permitted to participate in the final examinations of a subject if they have completed less than 80% of the required clinical hours, absenteeism significantly negatively influences student nurses [8]. Given the likelihood of having to repeat a level for a year, fewer students are graduating after four years, which contributes to the nation's shortage of qualified nurses. On the other hand, Ref. [2] argued that students' absence from the classroom and clinical environments prevented them from gaining the knowledge and skills they needed to attain their personal and professional goals. Furthermore, Ref. [2] showed that students' performance suffered when they were absent and their study time was extended. This additionally illustrates how little desire and to drive there is for studying. Similar to this, Ref. [14] demonstrated that student nurses' ability to provide high-quality care was negatively impacted by absence. According to Randa [2], student absenteeism results in poor skill application and prevents students from accessing the necessary knowledge and resources for optimal learning. Moreover, this is linked to substandard academic achievement, impolite behavior, and inadequate professional sociability. Hence, the present study sought the experiences of professional nurses regarding student nurses' absenteeism during psychiatric clinical placement in Limpopo Province, South Africa.

## 2. Methods

A qualitative approach was employed using explorative, descriptive, and contextual designs [15]. A qualitative approach was adopted because the study sought to express professional nurses' experiences of student nurses' absenteeism during psychiatric clinical placement in Limpopo Province, South Africa. The study also wanted professional nurses to describe their experiences regarding student nurses' absenteeism in clinical placement [16]. The study was contextual to the professional nurses working in specialized psychiatric hospitals of Limpopo Province only.

### 2.1. Study Setting and Population

The study was carried out at the three chosen specialized mental hospitals (Evuxakeni, Hayani, and Thabamoopo), which served as the clinical learning sites for nursing students at the Limpopo College of Nursing. The Greater Giyani sub-district is made up of rural areas with a population of many cultures, and the Evuxakeni specialist psychiatric hospital is located on the main road, Giyani, about 5.3 km outside the town. The three main language groupings are Sotho, Tsonga, and Venda. The Hayani specialist psychiatric hospital is located near Thohoyandou city in the rural Sibasa area along the main road. Those who speak Venda predominate there. About 50 km separate the Thabamoopo psychiatric hospital from Polokwane, located in the Lepelle Nkumpi local municipality in the South-Lebowakgomo East rural area. The local population converses in Sepedi. Eleven professional nurses participated in the study—four from hospital 1, four from hospital 2, and three from hospital 3. Three specialized psychiatric hospitals were included in the study as student nurses from Limpopo College of Nursing were placed there for their PNS clinical study. Only registered professional nurses who work in specialized psychiatric hospitals were recruited to participate in the study.

### 2.2. Sampling

Participants were visited in their respective hospitals to recruit them face-to-face a day before the interviews. Registered professional nurses who had supervised student nurses for three or more years from the selected hospitals in Limpopo Province were selected through purposive convenience sampling. Information regarding the study was explained briefly. Participants' consent was requested. The participants and researcher agreed upon the date, time, and venue for the interview. The researcher's contact numbers were given to the participants in case there were changes regarding the arrangements. Only participants who met the inclusion criteria were allowed to be part of the study.

### 2.3. Data Collection

Permission was sought from related authority bodies, namely the University of Venda Higher Degrees Committee, the Provincial Department of Health, the three specialized psychiatric hospitals, and the participants after approval and permission were granted.

The study's participants were made aware that they were not obligated to take part and could opt out at any point during the interview if they chose to. It was also explained that an audio recording would be used to record the conversation [16], and they were shown a stop button to stop the recording at any time if they wished to do so.

In-depth, semi-structured, one-to-one interviews [16] with registered professional nurses on duty during their lunch breaks were used to gather the data. The researcher encouraged participants to narrate their experiences by asking open-ended questions followed by probing questions. Each interview lasted for 35–45 min. Data were collected until saturation was reached after the ninth professional nurse from hospital number three was interviewed, as they gave out the same information. Nevertheless, the researcher continued with the interviews with the two remaining professional nurses to ensure that no new information was left out. The audio-recorded data were played back to participants to check for any missed information. The hospitals and participants were numbered from

Hospital 1 to 3 and Participant 1 to 11. The data were transcribed verbatim because the participants used English as a language.

*2.4. Data Analysis*

A tool developed by Tesch guided data analysis. Tesch provides eight steps outlined by [17] that should be considered when analysing qualitative data using a thematic approach. A coding scheme was used to identify the themes evident from the data by the panel of expert authors together with the principal author. The first author deducted the information from the collected data. A theme was allocated to a specific sub-theme that captured the same idea. The panel of expert authors scrutinized the deducted information and themes to increase validity. The authors then discussed the information and reached a consensus on the theme and sub-themes that emerged from the data, which were categorized accordingly. The theme was student factors leading to absenteeism, and the sub-themes were fear of mental health care users, negative attitudes towards psychiatric nursing science, and students' lifestyles.

*2.5. Ethical Consideration*

The Limpopo Province Department of Health, the CEO of the three chosen hospitals, the University Higher Degree Committee, the Research Ethics Committee of the University of Venda (SHS/20/PDC/04/1305) gave permission for the study, and participants signed consent forms before taking part. The environment where the interviews were conducted was safe for the participants since it was in the hospitals where they worked; therefore, no harm was done to the participants. The participants were reassured of confidentiality. An agreement with them was respected, including punctuality regarding the agreed time. Their names were not used: they were assigned numbers to ensure anonymity. COVID-19 protocols were also adhered to.

## 3. Results

*3.1. Demographic Data*

Eleven professional nurses participated in the study—four from each of the Evuxakeni and Hayani hospitals and three from the Thabamoopo hospital. Data were collected until saturation was reached in the interview with the ninth professional nurse. Nevertheless, the researcher continued with the planned interviews with the remaining two professional nurses to ensure that no new information was left out. Table 1 below indicates the demographic data of professional nurses.

**Table 1.** Demographic data.

| Hospital Number | Participant Number | Gender | | Work Experience |
|---|---|---|---|---|
| | | **Females** | **Males** | |
| No. 1 | P1 | | X | 11 YEARS |
| | P2 | | X | 8 YEARS |
| | P3 | X | | 14 YEARS |
| | P4 | X | | 24 YEARS |
| No. 2 | P1 | | X | 7 YEARS |
| | P2 | X | | 15 YEARS |
| | P3 | X | | 23 YEARS |
| | P4 | X | | 13 YEARS |
| No. 3 | P1 | X | | 12 YEARS |
| | P2 | X | | 5 YEARS |
| | P3 | X | | 16 YEARS |
| TOTAL | 11 | 8 = 72.7% | 3 = 27.3% | |

*3.2. Theme and Sub-Themes*

During data analysis, a theme emerged from participants' sharing of their experiences of the factors regarding student nurses' absenteeism. This theme comprised three sub-themes: fear of mental health care users, negative attitudes towards psychiatric nursing science, and students' lifestyles. This theme and the three sub-themes are discussed in detail below, with direct quotations from the transcripts. Each sub-theme is discussed separately.

Theme 1: Professional Nurses' Experience of Factors Regarding Student Nurses' Absenteeism

- Fear of mental health care users

Most professional nurses indicated that students were absent from psychiatric hospitals because of fear of mental healthcare users, particularly those students exposed to psychiatric wards for the first time. The following quotations indicate professional nurses' experiences of student nurses' fear of mental health care users.

One participant stated:

" ... I think they feel scared of patients thinking that they may do something bad to them because when some of the students go to them, it is like they are scared of them ... "

Participant 8C

Another participant said:

" ... Mmm in the hospital side, mmm some of them, especially the first level, some may give a reason that they are not comfortable working with mental healthcare users, those who are being exposed for the first time ... "

Participant 7B

Another participant mentioned that:

" ... And the behavior of the mental health care user towards the students. Is it indeed that, like those mental health care users, they spend most of their time in the ward, so they know that these are student nurses and the staff? You may find that how they treat students is unacceptable to such an extent that they become reluctant to go to work ... "

Participant 7A

- Lack of interest in psychiatric nursing science

Other professional nurses indicated that student nurses were negative toward psychiatric nursing science and were not interested in specific subjects while training. The following quotations indicate professional nurses' experiences of student nurses' lack of interest in psychiatric nursing science.

One participant said:

" ... They can be different; number 1, I can say students don't have an interest in what they are doing; maybe sometimes someone goes to Limpopo College of Nursing just because of the stipend; yah you find that he intended to do something in the University and he or she is not accepted there. So, he applied to Limpopo College of Nursing, and they took him, and I think that is the reason, lacking the interest ... "

Participant no 7A

Another participant said:

" ... Another thing is the attitudes of students towards psychiatry because you find that it is true that at the college we are being trained in different faculties, there are different subjects, so you find that a student has an attitude towards

a certain subject. Then you find that a student doesn't like psychiatry and tells himself that when I complete it, I don't want to work in psychiatry, so it makes him lose interest in psychiatry because he doesn't want to work in it on completing it also leads to absenting himself from work which is absenteeism . . . "

Participant no 8A

Another participant said:

" . . . The other thing, even themselves, I know the situation is boring, but even they don't have any interest; most are not. They just come and sit the whole day, no interviewing of the patients, nothing . . . "

Participant no 9B

Another participant said:

" . . . Another reason might be that they lack interest as students because some are not interested in the profession. You can hear from their conversation to finish and then work to accumulate funds so that we can start the business of our interests, so I think those are the reasons. Lack of supervision by professional nurses and lack of interest by students . . . "

Participant 9C

- Students' lifestyle

Some of the participants indicated that another reason why student nurses absented themselves from clinical areas might be their lifestyle, in particular issues relating to substance use, whereby students engage in intensive social activities for the whole weekend, drink alcohol and smoke, and wake up late on Monday, resulting in absenteeism. The following quotations indicate professional nurses' experiences of student nurses' lifestyle:

One participant mentioned that:

" . . . Yes, sometimes things like that may be part of the first thing I mentioned because you may find that a person is learning in nursing. On the other hand, he is drinking alcohol and smoking a cigarette like abusing substances, which can lead to absenteeism . . . "

Participant no 8B

Another participant said:

" . . . Another thing is that as student nurses, they are also parents from their families, so at their families, they may experience family problems at home. So, it may lead to absenteeism. Yes, social problems. You may find that they sometimes fight with their spouses and then experience communication breakdown, making them absent themselves . . . "

Participant no 9A

Another participant mentioned that:

" . . . When I look at it, I think it is because of their influence because when they come, the driver drops them at the gate, and they think nobody can see them, so maybe we don't have their schedule in the ward. So, they may sign, and we will think they are in the ward. It makes them absent themselves, thinking that we cannot see them as the people who accompany them, that is why they absent themselves . . . "

Participant 8D

## 4. Discussion

The findings of this study expanded knowledge of professional nurses' experiences regarding student nurses' absenteeism in specialized psychiatric hospitals. The findings of the present research, as well as those of some other studies, indicated that students

were absent from psychiatric hospital placement because of fear of mental healthcare users that made them stay away from the wards, particularly those students exposed to psychiatric wards for the first time. Strategies were developed in Iran [18] to lower student nurses' absenteeism, since interviewees there also raised concerns that people with mental problems were hazardous and destructive. Other students admitted to being afraid of suffering physical abuse at the hands of patients. However, to transfer teachers' knowledge, it is necessary that students are present in the classroom and clinical areas. In another study conducted by [19], the findings revealed that the negative view of people with mental illnesses as dangerous and harmful was raised by students during focus group discussions. Some students reported being terrified of being physically abused by the patients. Another study conducted in China by [20] revealed that only female nursing students were allowed to remain with the staff (midwives) and given the opportunity to learn, with male nursing students frequently being sent out of the patient's room. In addition, those subjected to this type of treatment felt hurt and isolated, wondering what they had done wrong to be treated in such a way. This could also cause student nurses to avoid clinical settings and develop a negative attitude toward the science of midwifery.

Other participants indicated that student nurses were negative toward psychiatric nursing science and were not interested in specific subjects while training. According to a different study by [21], the preparation for exams, a lack of interest, the teaching style of nurse educators, and the lecture content were the leading causes of absence. According to a study by [13], the information from interviews showed that some students enrolled in courses without thoroughly understanding the course material. They started to lose interest when they realized the course was not what they thought. In this situation, the importance of attending class is overlooked, resulting in a significant absence or dropout rate among the students.

In addition, some participants indicated that student nurses were absent from clinical areas because of lifestyle, social issues, and stresses they underwent as social human beings in the community. Participants also indicated that some students were married with children and had other family members to take care of, such as sick parents and siblings, which made them miss clinical attendance. This is in line with the findings of [22] that many students have family obligations that put additional demands on their time. Some students are in need of social assistance because they have young children and/or elderly parents with health issues. In addition, other factors include domestic disputes, family dissolution, divorce, childcare issues, troubled neighborhoods, and abuse. A student may become ill and miss class or clinical time. Our findings are similar to those of [10], that students miss class because they worry about getting paid to study but not paid to work. If they were paid to work, they could attend to family emergencies like sick children, spouses, or parents as needed. According to a study by [23], issues like physical illness, family obligations, and a lack of funds can all have an impact on student nurses' absences. The results show that there are a number of reasons why student nurses are not present in the clinical setting. Due to a lack of clinical hours, students are unable to get the skills required for patient care. In this study, professional nurses' experiences regarding student nurses' absenteeism from psychiatric clinical placements in Limpopo Province, South Africa, were examined. This goal was accomplished because when the researcher looked at professional nurses' experiences, they could describe and narrate their own experiences.

## 5. Limitations

This study was restricted to three of the five districts of the Limpopo Province. Furthermore, this study was contextual in that only professional nurses working in specialized psychiatric hospitals were interviewed; thus, the study could not obtain the perceptions of professional nurses working in the psychiatric wards of general hospitals where students from the Limpopo College of Nursing are also allocated. It is necessary to perform a qualitative study to determine strategies to mitigate student nurses' absences from their clinical placement in the psychiatric facilities of Limpopo Province.

## 6. Conclusions

The study's findings contribute to a better understanding of professional nurses' experiences of student nurses' absenteeism in psychiatric clinical placement. The study is important because the hospitals could benefit, as the study's findings may influence policymakers of the province for the betterment of patient care. The study also adds to the body of knowledge. It is relevant as it dealt with the experiences of professional nurses regarding student nurses' absenteeism in psychiatric hospitals but not in any other hospital.

**Author Contributions:** Conceptualization, T.C.M., M.M. and N.S.R.; Methodology, T.C.M., M.M. and N.S.R.; Formal analysis, T.C.M. and N.S.R.; Investigation, T.C.M., M.M. and N.S.R.; Resources, T.C.M., M.E.R., N.R.L. and T.E.T.; Data curation, T.C.M.; Writing—original draft preparation, T.C.M.; Writing—review and editing, M.M., N.S.R., N.R.L., D.M. Supervision, M.M. and N.S.R.; Project administration, T.C.M., M.E.R. and N.S.R. Funding acquisition, T.C.M. and N.S.R. All authors have read and agreed to the published version of the manuscript.

**Funding:** This research received no external funding.

**Institutional Review Board Statement:** The study was approved by the University of Venda Research Ethics Committee (SHS/20/PDC/04/1305).

**Informed Consent Statement:** Informed consent was obtained from all the participants involved in this study.

**Data Availability Statement:** The data supporting this study's findings are available from the corresponding author, T.C.M., upon reasonable request.

**Acknowledgments:** All authors would like to thank the University of Venda for the approval of the study and support, the Department of Health for allowing the study to be conducted, the CEOs of the three selected hospitals, and all participants of this study from the three selected hospitals.

**Conflicts of Interest:** The authors declare that no financial or personal relationships may have inappropriately influenced them in writing this article.

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
