# Peer review of "Professional Nurses’ Experiences of Student Nurses’ Absenteeism during Psychiatric Clinical Placement in Limpopo Province, South Africa: A Qualitative Study"

_nursrep, doi:10.3390/nursrep13020059_

Round 1

Reviewer 1 Report (Previous Reviewer 1)

Dear authors,

First, I would like to emphasize that the study is of great relevance in the construction of scientific knowledge and discusses in depth the absenteeism by students during their psychiatric clinical placement.

I believe that the study can contribute to other realities in the world.

As for requests and corrections, the authors did what was asked. Therefore, at this moment I consider that the article is ready for acceptance.

Author Response

Thank you very much for reviewing our manuscript and for the wonderful comments we received from you. Thank you.

Reviewer 2 Report (Previous Reviewer 2)

Thank you for submitting this manuscript for review. I am pleased to inform you that after careful consideration of this revisions and feedback, I find that this manuscript is now satisfactory, and I accept it for publication.

These revisions have greatly improved the clarity and coherence of their writing, and I appreciate the time and attention they have given to incorporating my feedback. This manuscript presents a valuable contribution to the field, and I am confident that their work will be of great interest to our readers.

Author Response

Thank you very much for reviewing our manuscript and for the positive feedback we received from you. Thank you.

Reviewer 3 Report (Previous Reviewer 3)

Thank you very much for the improvements made, the article better defines the objectives and purpose of the study.

You could improve some visual aspects, such as Table 3.1, (table 1: demographic data), and add (optional) a flowchart image of the methodology, it always helps to better understand the study.

Number well the table, title and this could be improved in terms of percentages and qualitative, quantitative data in the same.

thank you and good luck

Author Response

Thank you very much for reviewing our manuscript and for the positive feedback we received from you.

Attached receive the response to the comments. 

Thank you

This manuscript is a resubmission of an earlier submission. The following is a list of the peer review reports and author responses from that submission.

Round 1

Reviewer 1 Report

The manuscript is clear, relevant for the field and presented in a well-strutured manner. 

In abstract, it is recommended not to use reference (ex. [1].).

Considering the authors developed a qualitative study to investigate the absenteeism and its causes in student nurses, from the point of view of nurses. So, “absenteeism is caused by many factors” in abstract to “can be caused”.

To facilitate the coding of participants, it is suggested not to identify the hospital. For example, replace: Participant 8C – Hospital 2 with Participants 8C, regardless of hospital. Therefore, the objective was common to all and thus avoided the identification of the participant and the hospital. But, if the authors observed that there is a difference in the students' behavior depending on the Psychiatric clinical placement, it is suggested to work on this in the discussion.

About the references, just 9 of 20 references are within the last 5 years (> 2018).

Furthermore, the authors need to review the format of references. For example, all citations of articles published in journal need to show abbreviated Journal Name in references section.

Reviewer 2 Report

Thank you for the opportunity to review the manuscript. Overall, a special topic for concerning the development of nursing education and further exploration of this topic is certainly interest, especially to explore the experiences of professional nurses on student nurses’ absenteeism in Psychiatric clinical placement in South Africa.

A few questions / comments and suggestions:

In Materials and Methods, how to demonstrate the rigor of the study to COREQ checklist such as inclusion criteria of participants, what demographic data of participants such as any experiences in supervision clinical attachment for nursing students, their working experiences, etc.

In Line 134-153, how long to collect data and what time for the research team in recruitment and data collection, how to translate the verbatim in what language, relevant to the study is not clear.

In Line 194-252, I prefer authors demonstrate more fruity narrative from the participants among result findings.

In your discussion, one of the papers related to your research questions has not been reviewed to provide an in-depth comparison of findings across different context and timepoints. For example, this paper “Yip, Y. C., Yip, K. H., & Tsui, W. K. (2021). Exploring the gender-related perceptions of male nursing students in clinical placement in the Asian context: A qualitative study. Nursing Reports, 11(4), 881-890. doi: 10.3390/nursrep11040081” also relates to your study, but it was neglected. Consider acknowledging that paper and provide your interpretations/insights accordingly

In Line 254-256, relevant to the study is not clear.

In Line 273-274, ‘Participants also indicated that some are married with children and have other family members to take care of’, please rephrase the sentence and elaborate more, relevant to the study is not clear.

In Line 277-279, detail elaborate other factors, relevant to the study is not clear.

Suggested to add Subheadings of implication and limitation session for this manuscript.

Reviewer 3 Report

The following is the result of the evaluation.

The text is in many respects ambiguous, repetitive and lacking in precision.

The summary starts by talking about clinical practices, but these are just one of the specific practices, and the sentence does not make sense.

 In the qualitative approach, it begins by talking about eleven nurses, and tells us that four are from two specialised hospitals and three from others, which does not provide us with anything, (11) although later it does mention these eleven nurses.

He talks about the exploratory themes in the phenomenon and the qualitative methodology of the study (lack of information).

It does not specify how the sample was recruited, there are no inclusion criteria, the experience of these nurse tutors, the type of training, etc.

In general, the introduction jumps in time, and does not answer the question of why the work has been carried out, as it does not reflect or there is a lack of information about it, and the work is not put in the current context of the area of knowledge of psychiatry, nursing students, and the tutoring by the expert nurses.

The characteristics of patients in psychiatric institutions are not taken into account. The precise works are not cited or are not sufficiently clear.

Materials and methods are very generic in the design of the study and are repetitive in some aspects that are not important.

It talks about the fact that the nursing students carry out an experimental learning, but at no point does it relate it or in the introduction does it talk about the type of learning, its methodology, objectives or within the instrumental learning.

The inhabitants are important, but the study and its method is more important, that is why it is important to have a good methodological design.

It does not talk about the most prevalent problems, incidence of mental health diseases in the study area, there are no data.

Population of study: It speaks to us of these graduated professionals, but it does not create us and the content does not contribute and returns to repeat information in other previous sections.

Results: There is a lack of fundamental results of the study that clearly shows the main findings.

Discussion, the construction of the discourse is deficient with respect to what was stated in the introduction and according to the objectives and a coherent discussion with previous studies, as these would allow a discussion with these sources of discrepancy.

bibliography, I would make a new review
